# Genetic Adaptation by Dengue Virus Serotype 2 to Enhance Infection of *Aedes aegypti* Mosquito Midguts

**DOI:** 10.3390/v14071569

**Published:** 2022-07-19

**Authors:** Steven M. Erb, Siritorn Butrapet, John T. Roehrig, Claire Y.-H. Huang, Carol D. Blair

**Affiliations:** 1Center for Vector-Borne Infectious Diseases, Department of Microbiology, Immunology, and Pathology, Colorado State University, Fort Collins, CO 80523, USA; serb@rams.colostate.edu; 2Division of Vector-Borne Diseases, Centers for Disease Control and Prevention, Fort Collins, CO 80521, USA; sjb7@cdc.gov (S.B.); jtr1@cdc.gov (J.T.R.); yxh0@cdc.gov (C.Y.-H.H.)

**Keywords:** dengue viruses, *Aedes aegypti*, adaptive mutation

## Abstract

Dengue viruses (DENVs), serotypes 1–4, are arthropod-borne viruses transmitted to humans by mosquitoes, primarily Aedes aegypti. The transmission cycle begins when *Ae. aegypti* ingest blood from a viremic human and the virus infects midgut epithelial cells. In studying viruses derived from the DENV2 infectious clone 30P-NBX, we found that when the virus was delivered to female *Ae. aegypti* in an infectious blood meal, the midgut infection rate (MIR) was very low. To determine if adaptive mutations in the DENV2 envelope (E) glycoprotein could be induced to increase the MIR, we serially passed 30P-NBX in *Ae. aegypti* midguts. After four passages, a single, non-conservative mutation in E protein domain II (DII) nucleotide position 1300 became dominant, resulting in replacement of positively-charged amino acid lysine (K) at position 122 with negatively-charged glutamic acid (E; K122E) and a significantly-enhanced MIR. Site directed mutagenesis experiments showed that reducing the positive charge of this surface-exposed region of the E protein DII correlated with improved *Ae. aegypti* midgut infection.

## 1. Introduction

Dengue viruses (DENVs) (*Flaviviridae, Flavivirus*) are the most medically important arboviruses infecting humans today. It has been estimated that 390 million DENV infections occur annually, of which 96 million cause frank dengue disease [1] ranging from self-resolving dengue fever to life-threatening dengue hemorrhagic fever and dengue shock syndrome (DHF/DSS).

DENVs are maintained in nature in a mosquito-to-human transmission cycle. The primary mosquito vector, *Aedes aegypti,* is ubiquitously distributed throughout the tropics and is increasingly expanding beyond the tropics [2,3]. Vaccine development has been complex and problematic [4] and no specific anti-viral treatment is available. Thus, control of transmission by mosquito vectors is an integral facet in dengue disease reduction. Multiple steps are required for the transmission of mosquito-borne viruses to vertebrate hosts. Mosquitoes are exposed to the virus when ingesting an infectious blood-meal from a viremic host. The crucial initial step in the transmission cycle is infection of and replication within midgut epithelial cells. Although putative midgut cell receptor proteins have been shown to bind DENVs, they have not been characterized [5,6], and receptor-binding domain(s) on DENV structural proteins are not well-defined. After amplification and spread in infected midgut cells, virus must disseminate into the hemocoel to infect and further amplify in secondary target tissues including the salivary glands. The transmission cycle is completed when a salivary gland-infected mosquito inoculates a new host with virus during probing to seek her next blood-meal. The 10-to 14-day interval between initial infection of the midgut epithelium and ability of the mosquito to transmit the virus in the saliva is the extrinsic incubation period [7,8]. Four genetically and antigenically distinct DENV serotypes (DENV1-4) comprise the DENV serocomplex. Mature virus particles have a single-stranded positive-sense RNA genome (~11 kb) encapsidated in a protein core surrounded by a host-derived lipid bilayer anchoring 180 copies each of the virus-encoded membrane (M) protein and envelope (E) glycoprotein [9]. The E protein is arranged in an icosahedral scaffold of 90 homodimers that lie parallel to the virion surface. The flavivirus E protein, a Class II fusion protein, is responsible for host cell attachment and entry and virus envelope fusion with cellular endosomal membranes. The DENV2 E protein 2 Å crystal structure has been solved [10], revealing three distinct structural domains (DI, DII and DIII) that correlate with three previously described antigenic domains (C, B and A) defined by monoclonal antibody (MAb) mapping [11,12,13,14]. Domains I and II consist of linearly discontinuous amino acid sequences connected by four peptide strands that form a molecular hinge, which permits the translocation and insertion of the flavivirus-conserved DII fusion peptide into the host cell endosomal membrane to bring about virus-mediated cell membrane fusion [10]. DII also is involved in E protein homodimerization in the mature virion. DIII is an immunoglobulin-like structure connected to DI and has host-cell receptor binding properties [15].

Although it has been shown that different strains or genotypes of a single DENV serotype can exhibit differential specificity for infection of the same *Ae. aegypti* strain [16,17,18], to date there has been limited research focusing on DENV genetic determinants that influence infection of mosquitoes.

We have conducted studies to define the genetic determinants of the E gene/protein involved in infection of mammalian cells, using DENV2 strain 16681 derived from the infectious cDNA clone pD2/IC-30P-NBX (30P-NBX) [15,19,20]. We have also shown that DENV2 30P-NBX infects and replicates efficiently in various tissues outside the midgut of *Ae. aegypti* RexD strain mosquitoes after intrathoracic (IT) inoculation. However, infection of the midgut epithelium after introduction of virus by infectious blood-meal is inefficient [15]. The average midgut infection rate (MIR) for 30P-NBX in *Ae. aegypti* RexD in a large number of trials was 33.79%, compared to an average MIR of 78.74% for our standard laboratory strain DENV2 Jamaica 1409 (J1409) [21]. Given the importance of the E protein in determining DENV cell tropism/initiating infection, and that 30P-NBX has a relatively low infection rate after oral infectious challenge but not after IT inoculation, we hypothesized that mosquito midgut infection determinants are located in the E protein and that 16,681 had undergone mammalian cell culture-adaptive changes during multiple cell culture passages before preparation of the infectious cDNA clone that reduced its ability to infect midgut epithelial cells. Comparison of E protein AA sequences shows that there are nine differences between strains 16681 and J1409 and all except three of these, R120T, Q131L and T170I, are conservative. We noted that each time 30P-NBX was presented in a blood meal a small proportion of infectious clone-derived viruses efficiently initiated midgut infection, so in this study, we serially passaged 30P-NBX in *Ae. aegypti* RexD strain mosquitoes by oral infection to select for and identify E protein genetic adaptations conferring efficient midgut epithelial cell growth. We confirmed the importance of selected E protein domain II amino acid (AA) sequences in midgut infection by constructing and phenotyping 30P-NBX mutants.

## 2. Methods and Materials

### 2.1. Cell Culture and Viruses

Vero (African green monkey kidney epithelial) cells were grown at 37 °C in Dulbecco’s modified Eagle’s medium (DMEM) with 10% fetal bovine serum (FBS). C6/36 (*Aedes albopictus* larval) cells used for virus phenotyping experiments were grown in Ye-Lah medium [22] and C6/36 cells used for propagation of virus to infect mosquitoes were grown in Leibovitz L-15 infection medium containing 2% FBS, non-essential amino acids, penicillin-streptomycin, and L-glutamine, both at 28 °C. Both Vero cells and C6/36 cells are standard cell lines that have been used in our laboratories for a number of years. They were purchased from American Type Culture Collection (ATCC) (Vero CCL-81; C6/36 CRL-1660), and expanded by culturing in the media indicated. Aliquots of the expanded cultures were preserved in liquid nitrogen. After a given cell line had been recovered from liquid N_2_ and passed for experimental use no more than 20 times, it was discarded and replaced by a newly-recovered aliquot.

DENV2 strain 16681 was first isolated from the serum of a dengue hemorrhagic fever/shock syndrome (DHF/DSS) patient in Bangkok, Thailand, in 1964 [23]. The strain is a member of the DENV2 Asian 1 genotype and since isolation has been passaged multiple times in BS-C-1 cells, six times in LLC-MK2 cells, once in a rhesus macaque monkey, and twice in *Toxorhynchites amboinensis* mosquitoes [23,24]. The virus was additionally passaged once in Vero cells, twice in LLC-MK2 cells, and four times in C6/36 cells before construction of the infectious cDNA clone pD2/IC-30P-A [24]. The infectious cDNA clone was modified to facilitate introduction of site-specific mutations into the E gene and DENV2 derived from this infectious clone is termed 30P-NBX and exhibits the same phenotypes in mammalian and invertebrate cell cultures as parental strain 16681.

DENV2 strain Jamaica 1409 (J1409) was isolated from a human with dengue fever (DF) in 1983 and is a member of the American/Asian genotype. Strain J1409 was plaque-purified in LLC-MK2 cells and passaged extensively (>25 times) in C6/36 cells [25,26,27,28] before construction of the infectious cDNA clone (J1409-ic) by Pierro and colleagues [27]. After transfection and virus recovery in C6/36 cells, infectious clone-derived virus J1409 was passaged once in C6/36 cells for use in this study.

To introduce selected mutations into the E gene of infectious cDNA clone 30P-NBX, the QuikChange^®^ Lightning Site-directed Mutagenesis kit (Stratagene, Santa Clara, CA, USA) was used per the manufacturer’s instructions. Engineered mutations for this study were targeted to DII of the E protein to produce mutants KK122/123EE and R120T. Resulting recombinant cDNA was amplified and used as a template for transcription of infectious DENV2 genomic RNA. In vitro transcription for transfection of Vero and C6/36 cells was performed as described previously [19,22,24] using the AmpliScribe™ T7 kit (Epicentre Technologies, Madison, WI). Positive-sense vRNA was transfected into Vero and C6/36 cells using a Bio-Rad Gene Pulser Xcell system (Bio-Rad, Hercules, CA, USA) as described previously [19]. Medium harvested from transfected Vero cells (day 12 post infection (PI), designated V-0) or C6/36 cells (day 14 PI, designated C-0) was centrifuged to remove cell debris, supplemented with 20% FBS, and stored at −80 °C. Aliquots of V-0 and C-0 were used to infect naïve Vero and C6/36 cells to produce V-1 and C-1 seeds, respectively. Genome cDNA of V-1 and C-1 seeds were fully sequenced to evaluate their genomic stability as described previously [19].

### 2.2. Assay of Virus Growth Kinetics in Cell Culture

Twelve- to fourteen-day growth curves were performed to investigate mutant virus growth kinetics in various cell types. Duplicate cell cultures were infected with each virus at a multiplicity of infection (MOI) of 0.001. Virus genomic equivalents were measured by RT- quantitative (q)PCR using 3′-NCR primers and probes and the iScript™ One-Step RT-PCR Kit (Bio-Rad, Hercules, CA, USA) as previously described [29], and samples were assayed for infectious virus by TCID_50_ in C6/36 cells.

### 2.3. Infectious Virus Titration by 50% Tissue Culture Infectious Dose

Infectious virus end-point assays were performed in C6/36 cells with detection of virus antigen by ELISA as described previously [19]. C6/36 cells in 96-well plates were infected with 10-fold serial dilutions of virus and maintained for seven days at 28 °C with 5% CO_2_. Cells were fixed with cold acetone and virus antigen was detected by ELISA using DENV2 polyclonal antiserum. Absorbance was read at 405 nm and 630 nm and the difference was expressed as the delta optical density (ΔOD). ΔOD values 2-fold or greater than the negative control were considered positive. Virus titers were calculated by the method of Reed and Muench [30].

### 2.4. Aedes aegypti Mosquito Oral Infection by Artificial Blood-Meal

*Ae. aegypti* RexD strain mosquitoes (RexD) are a laboratory colony originating from Rexville, Puerto Rico. *Ae. aegypti* Chetumal strain mosquitoes are a more recently-established laboratory colony originating from Chetumal, Mexico [17]. They exhibit significantly higher midgut infection and dissemination rates for DENV2 J1409 than RexD mosquitoes [8]. All mosquitoes were reared from eggs and maintained as adults at 28 °C, 80% relative humidity with a photocycle of 12 h light: 12 h dark. Adult female mosquitoes were maintained in one-pint cartons with organdy covering and given water and sugar until infection. To prepare DENV2 for infectious blood-meals, C6/36 cells were infected at a multiplicity of infection (MOI) of 0.001 and maintained for 12–14 days with a medium change at 7 days. Infected cells were scraped into the medium, which was mixed with an equal volume of defibrinated sheep blood and supplemented with ATP to a final concentration of 1 mM. Adult female mosquitoes 4–6 days post-emergence were starved for 24 h, deprived of water for 4 h, and exposed to the infectious blood-meal for 45 min using a 37 °C water-jacketed glass feeding device with a hog-gut membrane. Virus titers in the blood-meals for all experiments ranged from 6.2 to 9.2 log_10_ TCID_50_/_mL_. Previous studies with 30P-NBX and E protein mutant viruses showed no correlation between virus titers in this range and midgut infectivity (data not shown). Fully engorged mosquitoes were selected and maintained for 7 days, when midguts were dissected in PBS, fixed in 4% paraformaldehyde in PBS overnight, and analyzed for virus antigen via immunofluorescence assay (IFA) to determine midgut infection rates. Each oral infection experiment was repeated at least three times with at least 19 mosquito midguts analyzed per experiment. To investigate mosquito infection rate kinetics, assays of dissected mosquito midguts and head tissues were performed every two days for 14 days post blood-meal (PBM). Virus antigen was detected in midguts and head squashes to determine infection and dissemination rates by IFA. Experiments were repeated three times and 17–30 mosquitoes were dissected at each time-point PBM.

### 2.5. DENV2 Passage in Aedes aegypti Mosquito Midguts

Infectious virus derived from DENV2 strain 16,681 infectious cDNA clone 30P-NBX, 30P-NBX-derived DIII FG loop mutants 382VEPGΔ and VEP382RGD (382RGD) [15], and DENV2 strain J1409 infectious cDNA clone (J1409) [27] were serially passaged in *Ae. aegypti* RexD midguts and amplified in C6/36 cells. To start the passage experiment, virus was amplified in C6/36 cells and an aliquot of the cell-virus suspension was incorporated into an infectious blood-meal as described above. The remaining cell-virus suspension was stored at −80 °C for titration and sequencing of the DENV E gene. RexD mosquitoes were challenged with the infectious blood-meal and fully engorged mosquitoes were maintained for 10 days. Mosquito midguts were dissected and placed into 4% paraformaldehyde for IFA analysis to determine MIRs (19–36 mosquitoes) or pooled on dry ice for trituration (at least 20 mosquitoes). Midguts were triturated in L15 infection medium and filtered through a 0.2 µm membrane syringe filter. The filtrate was placed directly onto naïve C6/36 cells to start the next passage. Four passages were completed for each virus and the complete passage series was repeated once for 30P-NBX.

### 2.6. Indirect Immunofluorescence Assay (IFA) of Mosquito Tissues

Midgut and head squash IFAs were performed as described previously [31]. Virus antigen in midguts and head tissues was detected using flavivirus E protein DII group-reactive mouse MAb 4G2 (HB-112, ATCC, Manassas, VA, USA) in wash buffer (PBS, 0.05% Triton X-100) or PBS, respectively. Secondary antibody was ImmunoPure biotin-labeled goat anti-mouse IgG (Thermo Scientific, Waltham, MA, USA) with 0.005% Evan’s Blue counter-stain, followed by streptavidin-fluorescein (GE Healthcare, United Kingdom). MIRs and head tissue infectivity (dissemination) rates were determined by dividing the number of virus antigen-positive midguts or head squashes by the total number analyzed. The relative infection intensity (RII) ratio is a quantitative measure of infection intensity in the midgut [31] in which positive midguts were scored on a scale of 0.5 to 4, where 0.5 denotes less than 25%, 1 denotes 25%, 2 denotes 50%, 3 denotes 75%, and 4 denotes 100% of the midgut surface area is positive for viral antigen. The RII ratio was determined by adding the infection intensity scores of all positive midguts in a treatment group and dividing by the total number of positive midguts. Student’s *t* tests (*p*-value ≤ 0.05) were performed using Excel 2007 and chi-square analysis (*p*-value ≤ 0.05) was carried out using SAS 9.1.

### 2.7. Envelope Glycoprotein Gene Sequencing

DENV2 RNA was isolated from infected C6/36 cell cultures with the QIAamp Viral RNA Isolation Kit (Qiagen, Germantown, MD, USA), and the E gene was amplified using the Titan One-Step RT-PCR system (Roche, Indianapolis, IN, USA) per the manufacturer’s instructions. PCR products were gel extracted using QIAquick Gel Extraction kit (Qiagen) and Sanger sequencing reactions were performed using ABI Prism BigDye Terminator v3.1 Cycle Sequencing Kit (Applied Biosystems, Carlsbad, CA, USA) at the Centers for Disease Control and Prevention, Fort Collins, CO. Sequences were analyzed using Lasergene Seqman (DNASTAR, Madison, WI, USA).

## 3. Results

### 3.1. Serial Passage of DENV2 in Aedes aegypti Mosquito Midguts

Infectious clone-derived DENV2 with four different E gene sequences were serially passed in *Ae. aegypti* mosquitoes to determine if adaptation to allow efficient midgut infection would result from the selection of virus genomes with enhancing mutations; the focus of our study was on the E gene due to its importance in initiating infection. DENV2 strain 16681 infectious cDNA clone-derived virus (30P-NBX); 30P-NBX-derived viruses with FG loop (DIII) deletion or mutation, 382VEPGΔ and VEP382RGD, which we had previously shown to decrease or not affect MIRs, respectively [15]; and DENV2 infectious clone J1409-ic-derived virus (J1409) were serially passaged by oral infection of *Ae. aegypti* RexD strain mosquitoes. Pre-passage virus MIRs of 30P-NBX and mutant virus VEP382RGD were similar, while mutant virus 382VEPGΔ had a significantly lower MIR and J1409 had a significantly higher MIR compared to 30P-NBX (Figure 1).

Each clone-derived virus was amplified in C6/36 cells, provided to RexD mosquitoes in an infectious blood-meal (SP0), and serially passaged four times (SP1-4) in mosquito midguts with amplification in C6/36 cells after each passage as described in Materials and Methods. Any increases in the rate of infection of midgut epithelial cells were assumed to result from mutations that occurred in the E gene during midgut infection since the E gene of 30P-NBX has been shown to be genetically stable during successive passages in C6/36 cells [29]. SP0 MIRs were similar to average MIRs previously determined for each virus (Table 1); 30P-NBX SP0 MIR fell within the range found in many previous determinations (data not shown). The MIRs of 30P-NBX and mutant VEP382RGD significantly increased after one passage in RexD midguts and in both cases, this increase correlated with a mixed nucleotide (nt) population of adenylic acid (A, parental) and guanylic acid (G, mutation) in consensus sequences at E gene position 1300. A transition from A to G causes a non-conservative AA change from lysine (K) to glutamic acid (E) in E protein DII at position 122 (K122E) (Table 1 and Figure 2). No other consistent nucleotide changes were seen in the remainder of E gene coding regions of genomes recovered from passaged virus. As revealed by consensus sequencing, four passages of 30P-NBX in mosquito midguts were required before E completely substituted K at position 122, while for mutant VEP382RGD, only two passages were required before this complete AA change occurred (Table 1). Two independent serial passage experiments were completed for 30P-NBX in RexD midguts and the second experiment yielded the same results (data not shown). In contrast, 382VEPGΔ and J1409 MIRs did not demonstrate any significant changes after four serial passages in mosquito midguts and no nt sequence changes were found in their E genes. Mutant 382VEPGΔ was lost after four passages and there was insufficient vRNA present in SP4 samples for sequencing. Our finding that a single AA mutation located in DII of the E protein appeared to significantly enhance mosquito midgut infection was unexpected.

### 3.2. Verification of Adaptation-Mutant Phenotype by Site-Directed Mutagenesis to Construct E Protein DII Mutant Viruses

The AA sequence of DENV2 16,681 (30P-NBX) E protein DII was aligned with those of other DENV2 genotypes, other DENV serotypes and other arthropod-borne flaviviruses. AA sequence alignments showed that there is considerable sequence conservation in DENV2, but sequence variability in other flavivirus DII AA 120 to 130 (Table 2). Comparison of E protein AA sequences revealed nine differences between strains 16,681 and J1409, only one of which occurs in this region. DENV2 strain 16,681 is the only flavivirus shown with positively-charged arginine (R) at position 120 All of the other DENV2 strains, including strain J1409, have uncharged threonine (T) at this position (Table 2). This difference is noteworthy due to the surface exposed location of T120 on the E protein and its close proximity to positively-charged KK122/123 (Figure 2). To determine if AA mutation K122E in DII of the DENV E protein was responsible for enhanced midgut infection, we introduced this and other potentially relevant mutations into the parental 30P-NBX genome by site-directed mutagenesis. After mutagenesis, RNA was transcribed in vitro and transfected into C6/36 cells (C-0) and virus produced was used to infect naïve C6/36 cells (C-1). Mutant KK122/123EE, which had previously been developed, was able to replicate after both transfection and one passage in C6/36 cells as evidenced by detectable viral antigen in cells. Consensus sequencing of the C-1 virus genome showed the expected full length genome sequence (Table 3), demonstrating that these mutations had no effect on infectivity of or replication in C6/36 cells. Mutant virus RNA also was transfected into Vero cells (V-0) and resulting virus was used to infect Vero cells (V-1). Full-length genome sequencing of KK122/123EE V-0 virus showed a partial reversion from E at position 123 to K, while E at position 122 remained unchanged. V-1 genome consensus sequencing revealed a complete reversion to K at position 123 while E at position 122 remained unchanged (Table 3). Recovered V-1 virus contained the mutation selected in *Ae. aegypti* midgut serial passage experiments with no other differences in the virus genome sequence compared to 30P-NBX (K122E), and was used as virus seed in all subsequent phenotypic studies. Virus recovery and genome sequencing showed that K123 was essential for replication in Vero cells, although either K123 or E123 was tolerated in C6/36 cells.

Interestingly, J1409 (T120, KK122/123) did not accrue any adaptive mutations during passage in mosquito midguts. Site-directed mutagenesis was used to introduce mutation R120T into the E protein of 30P-NBX (mutant designated R120T). Mutant R120T replicated in both C6/36 cells and Vero cells without alterations in genome sequence after both transfection and one virus passage (Table 3).

Growth kinetics of parental and mutant viruses K122E and KK122/123EE were analyzed by infecting duplicate cell cultures with each virus at a MOI of 0.001 and measuring virus genomic equivalents in medium by RT-qPCR or infectious virus by TCID_50_, as described in Methods, every two days. Growth kinetics for both mutants were similar to 30P-NBX in C6/36 cells (Figure 3A), corroborating the transfection data and the ability of these viruses to replicate efficiently in this cell type. Interestingly, 30P-NBX consistently caused more cytopathic effects (CPE) in the form of syncytium formation, cell rounding, and cell detachment than K122E in the C6/36 cells routinely used to amplify virus for oral infection experiments (data not shown). In Vero cells grown at 37 °C, mutant K122E replicated similarly to 30P-NBX and had equivalent virus genome titers, in contrast to mutant KK122/123EE, which showed no virus RNA replication (Figure 3B). Sequencing virus RNA at the conclusion of the growth curves showed K122E was genetically stable, while there was not enough KK122/123EE viral RNA recovered from Vero cell medium for sequencing at the conclusion of the experiment. Growth kinetics of mutant R120T in Vero and C6/36 cells were determined by titrating infectious virus released into medium by TCID_50_ (Figure 3C,D). R120T reached similar peak titers as 30P-NBX in both cell types, but peak titer was reached more rapidly than 30P-NBX in Vero cells (Figure 3D). Due to the apparent inability of mutant KK122/123EE to replicate at 37 °C, its temperature sensitivity was investigated by comparison of growth in Vero cells at 28 °C and 37 °C (Figure 3E). No apparent virus replication occurred at 37 °C up to day 2 PI in this experiment, as seen previously (Figure 3B). However, at 2 days PI, KK122/123EE began to replicate at a more rapid rate at 37 than at 28 °C, and this continued from days 4–8 PI. After day 8, the rate of virus replication became similar at the two temperatures and virus genome titers were similar by day 12 PI at both temperatures. Virus genome sequencing revealed no additional nucleotide changes in mutant KK122/123EE after replication at 28 °C, while mutant KK122/123EE grown at 37 °C partially or fully (results from duplicate cultures) reverted from E123 to K123. We attribute the difference from results shown in Figure 3B to a random mutation that changed E123 to K123 and was thereafter selected for more rapid growth. These results suggest that initial mutation is a stochastic event, followed by selection of mutants with a growth advantage. The results shown in Figure 3B,E suggested that lysines at both positions 122 and 123 of the E protein are not tolerated for replication at 37 °C. Interestingly, reversion of only AA 123 and not AA 122 occurred in the transfection, passage, and growth curve assays.

### 3.3. Phenotypic Properties of E Protein DII Mutants in Ae. aegypti Mosquitoes

To verify that mutation K122E in E protein DII, which was selected during 30P-NBX serial passage in *Ae. aegypti* midguts, was responsible for enhanced midgut infection and replication, constructed mutants K122E as well as KK122/123EE and R120T were presented to adult female mosquitoes in infectious blood-meals and MIRs were determined at 7 DPM. Mutant viruses each had significantly higher MIRs [K122E (81.8%), KK122/123EE (81.9%) and R120T (79.6%)] than 30P-NBX (33.8%) in RexD mosquitoes. (Figure 4A) Complete genome sequences showed that the only difference in deduced AA sequences between K122E and 30P-NBX recovered from mosquitoes was at position 122, confirming that this mutation alone was responsible for the enhanced infection rates that developed during serial passage. Significantly increased MIRs of mutants KK122/123EE and R120T than parental 30P-NBX implicated reduction in surface-exposed, positively charged AAs in E protein DII in increased efficiency of midgut infection (see Figure 2). To determine if 30P-NBX MIRs were higher in a more susceptible *Ae. aegypti* strain, mosquitoes from the Chetumal colony were provided with infectious blood-meals containing 30P-NBX and mutants K122E and R120T. Although 30P-NBX had higher MIRs (38.85% compared to 33.79% in RexD), K122E and R120T also had significantly higher MIRs compared to 30P-NBX in Chetumal mosquitoes (Figure 4B). Four independent challenge experiments were conducted, each completed at least one month apart. Completing all experiments concurrently would have utilized mosquito eggs that were oviposited by the same parents. Also, all of the extrinsic environmental factors would have been similar for each of the repetitions, which might further bias our results. Conducting replicate experiments at different times reproduced our previously observed general experimental variation in MIRs by 30P-NBX and other DENV2 strains, as seen in Table 1.

IFA analysis of infected mosquito midguts to determine MIRs also showed that on average, in RexD infections by K122E, KK122/123EE, and R120T, DENV2 E antigen was detected over a significantly greater area of the entire midgut at all times PBM than in mosquitoes infected with 30P-NBX. To express this difference, a measurement of relative midgut infection intensity, the RII ratio, was developed as described in the Methods section and illustrated in Figure 5. Mutants consistently had higher RIIs than 30P-NBX (Figure 5) Our observations suggested that K122E initiates infection in a higher proportion of cells and spreads more rapidly and efficiently in the mosquito midgut than 30P-NBX.

To determine if the more efficient and robust midgut infection by mutant K122E resulted in a higher infection rate in secondary mosquito tissues, implying a higher transmission rate, virus dissemination from the midgut was investigated. IFA analysis for DENV2 E antigen in head tissues from mosquitoes in which MIR and RII had been determined showed that both K122E and 30P-NBX began to escape the midgut at four days PBM. However, K122E had a significantly higher infection rate in head tissues than 30P-NBX by day six PBM, continuing until the end of the time course (observations not shown). Thus, mutant K122E both infects and disseminates from a higher proportion of mosquito midguts than 30P-NBX.

## 4. Discussion

In this study we demonstrated that single amino acid replacements/changes in DII of the DENV2 E protein of a virus with low MIR can result in significantly enhanced infection of *Ae. aegypti* mosquito midguts. Serial passage of 30P-NBX in RexD mosquitoes identified an adaptive mutation in DII of the E protein at position 122 from positively-charged lysine to negatively-charged glutamic acid (K122E) that correlated with increased infection rates in mosquito midguts. Incorporation of this mutation into the infectious clone recapitulated the results of the serial passage experiment, showing that this single AA mutation was solely responsible for the enhanced infectivity phenotype. We also showed that an alternative single mutation of spatially proximate positively-charged AA R120 to uncharged T significantly enhanced mosquito midgut infection compared to the parent virus. To our knowledge this is the first time mosquito infection determinants have been mapped to DII of the DENV E protein.

Time course experiments in mosquitoes showed that mutant K122E initiated infection in a significantly higher proportion of mosquitoes than 30P-NBX as early as two days post blood-meal (PBM), suggesting that early stage events such as attachment and/or entry were enhanced by the K122E mutation. The specific cellular receptor(s) for DENVs in either human or mosquito target tissues and receptor-binding domains on DENV structural proteins for these receptor(s) is/are unknown. It is possible that mutant K122E and 30P-NBX have different attachment affinities for a specific primary or ancillary receptor, thus contributing to differences in infectivity rates. In addition, mosquitoes were challenged with virus that was maintained at 37 °C and this higher temperature might have resulted in virion structural/conformational changes [32] that promoted enhanced receptor affinity in mutants K122E and R120T [33], as well as accelerated attachment and entry kinetics in the mosquito midgut that affected MIRs. Maintaining the blood-meal at a temperature of 28 °C or lower may help elucidate whether temperature differentially affects receptor-binding affinities or internalization of the two viruses.

DIII of the flavivirus E protein is widely accepted to have receptor binding properties and the FG loop (AAs 381–386) specifically was proposed to bind to mosquito cells. This was further suggested by the absence of this loop structure in the tick-borne viruses. Previously we showed that deletion of the FG loop AAs 382–385 (VEPG) attenuated virus infection in mosquito midguts as well as in Vero cells. Mutation of the FG loop AA sequence 382–384 from VEP to RGD did not significantly affect MIRs, suggesting that the FG loop structure itself and not the AA sequence is important for midgut infection [15]. Due to our focus on E protein functions required for midgut infection in this study, we included mutant viruses 382VEPGΔ and VEP382RGD in the serial passage experiments to determine if deletion or alteration of this DIII motif would place selective pressure on the E protein to acquire mutations that would compensate for these changes. Serial passage of 382VEPGΔ in mosquito midguts did not select for any enhancing adaptive mutations and the virus was lost between the third and fourth passages. Although this virus was able to infect mosquito midguts and secondary tissues in the first passage (albeit at a significantly lower rate than 30P-NBX), our inability to continuously passage 382VEPGΔ in midguts indicated that the presence of the FG loop is vital to the transmission cycle of the virus in vivo and no compensatory changes in other domains could rescue its loss. In contrast, mutant VEP382RGD acquired the K122E mutation more rapidly than 30P-NBX, after only two passages in mosquito midguts. This may suggest that the RGD substitutions imposed greater selective pressure for the K122E mutation. Additionally, multiple passages of wild type DENV2 strain 16681 in Vero cells resulted in a mixed K122K/E population (C.Y-H. Huang, unpublished data). These findings at first suggested that replacement of a positive charge at AA 122 was primarily a primate cell culture-adaptive mutation for DENV2, but the results of this study show the K122E mutation is also relevant to invertebrate systems. Whatever the selective pressure, it appears that reduction of the number of positively charged AAs in this region of DII constitutes a mutational hot spot in the E protein of DENV2 strain 16681.

The high MIR of J1409 may have precluded selective pressure for this DENV2 strain to accumulate mutations during passage in mosquito midguts. DENV2 E protein AA sequence alignments showed that strain 16681 has R120 while all other DENV2 strains, including J1409, have T120 (Table 3). The close proximity of AA120 to AA122 (Figure 2) suggests that replacement of a positively-charged AA at position 120 may have a similar effect to K122E and indeed, mutant R120T had a significantly higher MIR than 30P-NBX, showing that this AA substitution alone could also result in the increased midgut infectivity phenotype. Strain 16,681 has been extensively passaged in various mammalian and invertebrate systems since isolation and the positively-charged R120 could be the result of those passages. This is in agreement with findings that showed passage of DENV2 strain PUO-218 in cultured mammalian BHK-21 cells selected for a T120K change that resulted in higher binding capacity for glycosaminoglycans (GAGs) and reduced neurovirulence in mice [34]. The high MIR resulting from mutation R120T in 30P-NBX may explain why J1409 did not accrue any adaptive mutations in the E gene during midgut passage and similarly, the absence of R120 in other DENV2 strains may explain why E122 is not present in any natural isolates. A reduction in total surface-exposed, positively-charged AAs in this region of E protein DII appears to be a critical requirement for mosquito midgut infection, thus accounting for low MIRs for 30P-NBX, with R120/K122/K123 in its DII sequence, in even highly susceptible Chetumal strain mosquitoes. Reduced DII positive charge density is also tolerated in mammalian cells, as shown by the ability of K122E to replicate in Vero cells as efficiently as 30P-NBX. Nevertheless, engineering double mutations K122E and K123E while retaining R120T rendered the virus unstable in Vero cells at 37 °C (Figure 3B and Figure 4). Interestingly, the double mutant was able to infect a significantly higher proportion of mosquito midguts than 30P-NBX, suggesting that KK122/123EE facilitates infection and is stable during replication in midgut cells at 28 °C.

Reduction of positive charge density on the E protein DII surface also could alter E protein monomer or homodimer stability. For example, two phenotypically distinct DENV2s with an E protein AA difference at position 62 were isolated from K562 and C6/36 cells inoculated with serum from the same DHF patient [35]. The virus isolated from C6/36 cells had E62 and could not bind to and infect B lymphocytes, whereas virus isolated from K562 cells had K62 and was able to bind to and efficiently infect B cells. The K562 cell-derived virus was capable of infecting C6/36 cells but with low efficiency. AA 62 is located in close proximity to AAs 122 and 123 at the homodimer interface (Figure 2) and Kinoshita et al. [35] speculated that K (compared to E) at this position causes high electrostatic repulsive effects with its sister AA on the opposite monomer, causing a loosening effect favorable to B cell binding. However, K122, K123, and R120 do not appear to have such close proximity to their equivalent AAs on the opposite monomer (Figure 2) so it is less likely that electrostatic repulsion affects these AAs to the same extent.

It also is possible that E protein DII mutations affect conformation and stability during virion assembly and maturation. DENV midgut infections generally start at a few foci of infection and spread laterally from each infected cell either by direct cell-to-cell transfer at the edges of a focus or by budding out of cells and diffusing short distances to infect local cells [8]. RII ratios in the midgut for mutant K122E were significantly higher than for 30P-NBX as early as day four PBM, showing that K122E causes an extremely productive infection that spreads rapidly and eventually encompasses the entire tissue. 30P-NBX midgut infections remain relatively restricted by comparison. Mutations on the exposed surface of DII may stabilize the E protein during assembly and maturation through the trans-Golgi pathway, helping to produce a higher infectious virus to particle ratio. Prestwood et al. [34] suggested that the N124D/K128E mutant virus produced a higher PFU to particle ratio than the parent virus in BHK-21 cells. Perhaps a similar phenomenon occurs in mosquito midgut cells.

Although the most potent and serotype-specific neutralizing anti-DENV monoclonal antibodies (MAbs) bind to epitopes on E protein DIII, both murine and human neutralizing MAbs have been mapped to DII epitopes [36,37]). The neutralizing MAb-binding region of DII is variable among the DENV serotypes and other flaviviruses (Table 3) and AA mutations in this region have been shown to create flavivirus MAb neutralization escape variants for DENV, JE, MVE, TBE, and YFV (summarized in [38]), suggesting that the AA sequence variability among flaviviruses in this region of the E protein may be reflective of enhanced immune pressure in the vertebrate host.

DENV2 are suggested to have evolved only recently from sylvatic DENV2 maintained in nature in a cycle between non-human primates and canopy-dwelling *Aedes* spp. Mosquitoes [39,40]. Phylogenetic analysis of DENV2 E proteins suggested that a L122K mutation (Table 3) was predicted to have accompanied DENV2 E protein evolution to the *Ae. aegypti*-human cycle, possibly due to immune selection [38,39,40,41]. Most of the AA mutations proposed to correlate with emergence of endemic/epidemic DENV1-4 from sylvatic progenitors were located in DIII, so it is unclear from the phylogenetic analysis whether the L122K change occurred independently or in combination with changes in DIII. If this mutation was necessary for the transition from the sylvatic to the human cycle, it could explain why no natural DENV2 endemic/epidemic isolates have mutations of K122.

The epidemiological implications of our findings are highlighted by the dissemination data from the time course experiment. Head tissue IFA analysis, used as a surrogate for transmission potential, showed that K122E disseminated from the mosquito midgut in a higher proportion of mosquitoes than in 30P-NBX. Even though this may correlate more with higher MIRs and/or higher proportion of infected midgut cells (RII) than with increased dissemination capacity over 30P-NBX virus, the ability of this mutant virus to disseminate from the midgut and infect secondary tissues is clearly greater than 30P-NBX. It will be interesting to see if mutant K122E can infect field-caught or genetically diverse laboratory strain *Ae. aegypti* mosquitoes as efficiently as the laboratory colonized strains used in this study. Increases in viral fitness that increase transmission rates by mosquitoes and produce higher viremia titers in humans can lead to genotype and strain displacements [42,43,44,45]. Given that the K122E mutation does not result in fitness costs for replication in mammalian cells, the epidemic potential of a virus that accumulates this point mutation would likely be high.

We have shown for the first time that single AA mutations in DII of the DENV2 E protein can significantly enhance infection of *Ae. aegypti* mosquitoes. Natural mosquito isolates of DENV2 should be monitored for variations in gene sequence at this surface-exposed region of DII and could provide biological markers for virus emergence in the future. Inclusion of arginine at position 120 in DII of live-attenuated DENV2 vaccine viruses with the predominant KK122-123 sequence in addition to attenuating mutations could reduce the transmission potential of vaccine viruses from vaccinees. Investigation of the contribution of this AA region to protective immunity, DENV transmission, and viral pathogenesis in mammals merits further attention.

## Figures and Tables

**Figure 1 viruses-14-01569-f001:**
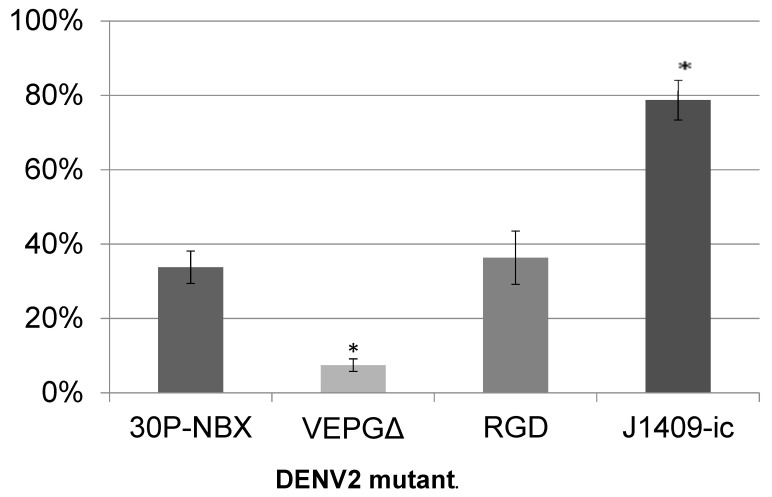
DENV2 MIRs in *Ae. aegypti* RexD mosquito midguts. RexD mosquitoes were orally challenged with each virus, maintained for seven days until midguts were dissected, and MIRs were determined by IFA. Data are the average of at least three experiments and significance was determined by comparison with 30P-NBX via student’s *t* test (* *p*-value < 0.05).

**Figure 2 viruses-14-01569-f002:**
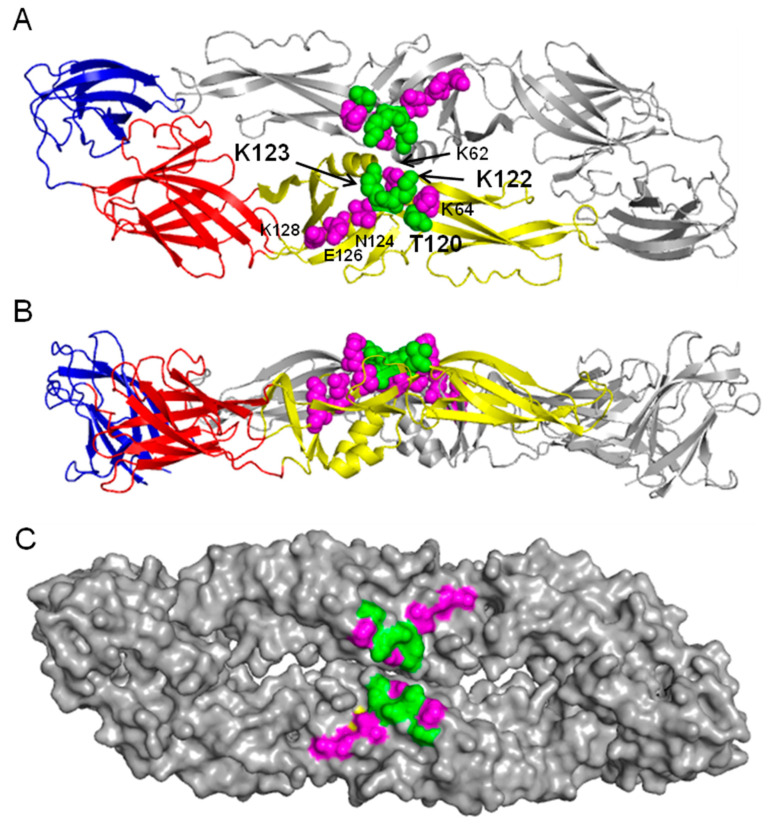
DENV2 E protein structure and location of relevant AAs in DII. (**A**) Top-down view of the DENV2 E protein homodimer with DI in red, DII in yellow, and DIII in blue in lower left monomer. AAs examined (T120, K122, and K123 in green) and discussed (K63, K64, N124, E126, and K128 in magenta) in this study are specified. Note, AA 120 is threonine in the published DENV2 E protein structure [10]. (**B**) Side-view of the DENV2 E protein homodimer. (**C**) Top-down view of the space-filling model of the DENV2 E protein homodimer to show surface-exposed AAs, which are colored the same as in A. Protein structures were obtained from the protein database bank (DENV2 E protein homodimer ID: 1oan) and were rendered in Polyview-3D.

**Figure 3 viruses-14-01569-f003:**
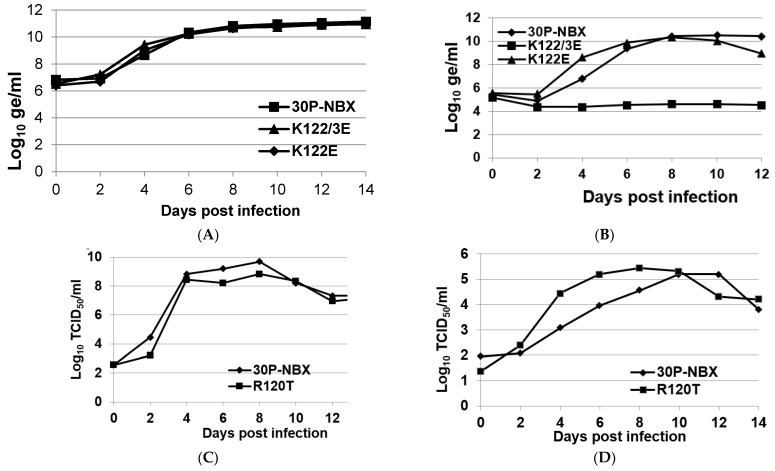
Virus growth kinetics in C6/36 and Vero cells. C6/36 cell cultures at 28 °C (**A**) and Vero cell cultures at 37 °C (**B**) were infected with DENV2s at an MOI of 0.001 and medium was sampled every two days for 14 or 12 days. Data presented are geometric mean titers (in log_10_ genome equivalents/mL) from duplicate flasks determined by RT-qPCR. C6/36 cell cultures at 28 °C (**C**) and Vero cell cultures at 37 °C (**D**) were infected with 30P-NBX and R120T at a MOI of 0.001 and infectious virus in medium was measured by TCID_50_ every two days for 14 days. The data presented are geometric means (in log_10_ TCID_50_/_mL_) from duplicate flasks. (**E**) Duplicate Vero cell flasks were infected with mutant KK122/123EE and maintained at 37 °C or 28 °C. Titers were determined by RT-qPCR. Virus RNA was sequenced at the end of the growth curve to verify the status of engineered mutations. Mutant KK122/123EE retained the mutated sequence at 28 °C but the sequence partially or fully reverted to E123 at 37 °C.

**Figure 4 viruses-14-01569-f004:**
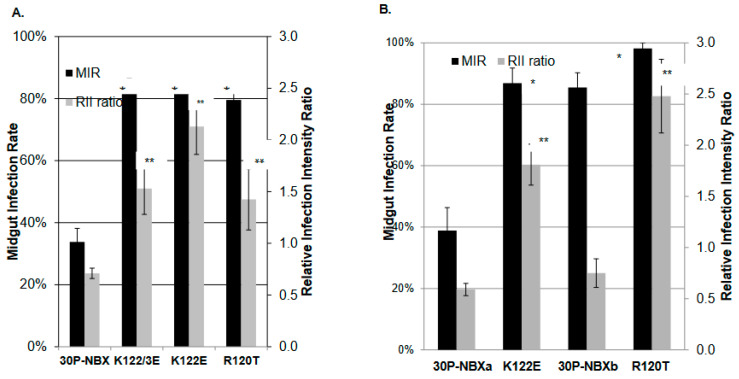
Virus MIRs and relative infection intensity (RII) ratios in Ae. aegypti RexD mosquito midguts (**A**) and Chetumal mosquito midguts. Mosquitoes were orally challenged with each virus, maintained for seven days until midguts were dissected, and MIRs and RII ratios were determined by IFA. In (**B**), 30p-NBXa and 30P-XNXb are data for internal controls for K122E and R120T, respectively. Data are the average of at least three experiments and significance of MIR (*) and RII ratio (**) differences were determined by comparison with 30P-NBX using student’s *t* test (*p* value < 0.05).

**Figure 5 viruses-14-01569-f005:**
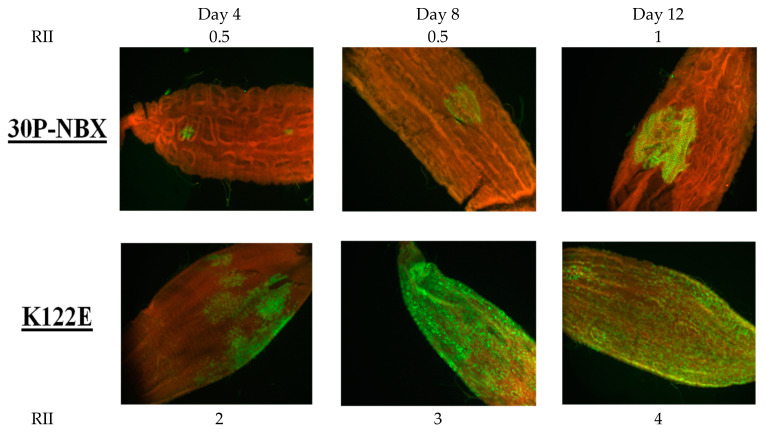
Demonstration of RII ratios in infected midguts from the time course experiment. RexD mosquitoes were orally challenged with 30P-NBX or K122E, midguts were dissected every other day for 14 days, and virus antigen was detected by IFA. Proportions of midguts displaying virus antigen (green) are representative of typical infection intensity seen for day 4 (**left**), day 8 (**middle**), and day 12 (**right**) PBM.

**Table 1 viruses-14-01569-t001:** Midgut infection rates and genetic changes after serial passage of 30P-NBX, 382VEPGΔ, VEP382RGD, and J1409 in *Ae. aegypti* RexD mosquitoes.

	30P-NBX			382VEPGΔ			VEP382RGD			J1409		
Passage	Midgut Infection Rate (%) ^a^	Confi-dence Interval (%)	Position 122 AA ^b^	Midgut Infection Rate (%) ^a^	Confi-dence Interval (%)	Position 122 AA ^b^	Midgut Infection Rate (%) ^a^	Confi-dence Interval (%)	Position 122 AA ^b^	Midgut Infection Rate (%) ^a^	Confi-dence Interval (%)	Position 122 AA ^a^
0	6/34 (18)	7–34	K	1/24 (4)	1–20	K	11/25 (33)	20–51	K	17/19 (89)	68–97	K
1	13/19 (68)	46–86	K/E	5/31 (16)	7–33	K	25/30 (83)	66–93	K/E	15/28 (54)	36–71	K
2	22/36 (61)	45–75	E/K	4/34 (12)	5–27	K	27/29 (93)	78–98	E	22/35 (63)	46–77	K
3	15/34 (44)	29–61	E/K	1/27 (4)	0.9–18	K	12/21 (57)	36–76	E	24/31 (77)	60–89	K
4	18/19 (95)	75–99	E	0/35 (0)	0.07–10	NA ^c^	26/30 (87)	70–95	E	25/29 (86)	69–94	K

^a^ MIR determined by dissecting and staining midguts with E protein-specific fluorescent antibody at 7 days PBM. ^b^ AA present at DENV2 E protein position 122. Mixed consensus cDNA sequence determined for both strands. Codon for the first AA is present in greater amounts than the second AA, as determined by examination of the sequence chromatogram. ^c^ Not available. There was insufficient vRNA present for sequencing.

**Table 2 viruses-14-01569-t002:** E protein DII AA sequence alignment of DENV2 and other flaviviruses.

Virus	Strain	DENV2 E Glycoprotein AA Position																
		116	117	118	119	120	121	122	123			124	125	126	127	128	129	130
**DENV2 Genotypes**																		
Asian 1	**16681**	C	A	M	F	**R**	C	**K**	**K**	- ^a^	- ^a^	N	M	E	G	**K**	V	V
	PUO-218					T												
	M1					T											I	
Asian 2	New Guin C					T								K				
	PL046					T											I	
	CTD113					T								K				
Asian/Amer	Jamaica 1409					T												
	13382-Tizimin					T												
American	PR159					T											I	
	Ven2					T											I	
Cosmo	SL714					T											I	
	CAMR5					T											I	
Sylvatic	IC80-DAKAr578					T		L				K						
	P8-1407					T		L										
**Other Mosquito-Borne Flaviviruses**																		
DENV1	16007			K		K		V	T			K	L				I	
DENV3	PhMH-J1-97			K		Q		L	E			S	I					
DENV4	Thailand/1985			K		S		S	G			K	I	T		N	L	
YFV	Asibi			K				A				S		S	L	F	E	
JEV	Nakayama			K		S		T	S			K	A	I		R	T	I
MVEV	NG156			K		T		S	S			S	A	A		R	L	I
WNV	NY99			K		A		S	T			K	A	I		R	T	I
SLEV	Laderle			K					N			K	A	T			T	!
**Tick-Borne Flaviviruses**																		
TBEV	Neudoerfl		V	K	A	A		E	A	K	K	K	A	T		H		Y
POWV	LB			K		E		E	E	A	K	K	A	V		H		Y

Positively-charged AAs in DENV2 strain 16681 sequence are in bold. Blank spaces indicate AA is identical to that in DENV2 strain 16681. ^a^ Tick-borne flaviviruses have 2 AA inserted between positions 123–124 of mosquito-borne viruses.

**Table 3 viruses-14-01569-t003:** Transfection and infection by 30P-NBX E protein DII mutants in C6/36 or Vero cells.

Virus Mutant	C6/36 Cells		Vero Cells		
	C-0/C-1 ^a^ Virus Recovery	C-0/C-1/E Protein Sequence	V-0/V-1 ^a^ Virus Recovery	V-0 E Protein Sequence	V-1 E Protein Sequence
KK122/123EE	+/+	Unchanged ^b^	+/+	K122 E ^c^, K123K/E part. rev. ^c^	K122 E ^c^, E123K full rev. ^c^
R120T	+/+	Unchanged ^b^	+/+	Unchanged ^b^	Unchanged ^b^

^a^ Transfection (C-0 or V-0) and recovered virus passage (C-1 or V-1) was considered positive if virus antigen was detected in cells by IFA. ^b^ Sequencing verified that recovered virus genomes contained the introduced mutations and had no additional changes in the genome. ^c^ Mutants with partial reversions (part. rev.) or full reversions (full rev.) in the E protein gene are specified.

## Data Availability

Not applicable.

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
