# Peer review of "Genetic Adaptation by Dengue Virus Serotype 2 to Enhance Infection of Aedes aegypti Mosquito Midguts"

_viruses, 2022, doi:10.3390/v14071569_

Round 1
Reviewer 1 Report
Viruses-1794808
The current study was designed to test if an infectious DENV clone could be optimized to infect Ae. aegypti midgut tissue, which could identify key residues that modify virus transmission. Multiple cDNA clones were passaged by cycling between C6/36 cells and midgut tissue. A specific amino acid in domain II of E was selected in a few clones, which correlated with an enhanced midgut infection rate. Although this residue was not present in published sequences, a neighboring charged to uncharged mutation recapitulated the phenotype. The result of this study may be an artifact of the hybrid in vitro/in vivo system that was used, although it is a solid demonstration of how we can leverage natural selection to understand how viruses may adapt in the wild.
Major concerns:
- No major concerns.
Minor concerns:
- Y-axis is not labeled in Figure 1.
- X-axis inconsistently labeled in Figure 3.
Author Response
Thank you for your helpful comments. I have attempted to consistently label the figures. Unfortunately, when the editors converted our manuscript and figures to their preferred format, some of the figure labels were lost.
Reviewer 2 Report
In this manuscript, the authors report on point mutations in domain II of the Dengue 2 virus E protein that enhance infection of mosquito midguts and possibly mosquito transmission. The manuscript is well written and the results support the conclusion for the most part. Some items to consider to improve clarity of results:
In the paragraph on Growth Kinetics in vitro (322 to 351) and Figure 3. Please justify and discuss why TCID50 was used instead of GE/ml when assessing R120T mutants. Also, the difference between B and E needs better explanation. Particularly with respect to why the mutant didn't grow in the first experiment at 37C (B) as compared to the second experiment (E). How many replicates were performed? Where are the error bars? Were any statistics performed to assess differences in growth kinetics?
Figure 4 figure legend indicates that significance was assessed using student's t test, but neither the figure or the discussion indicates that results of this analysis.
Author Response
Thank you for your helpful comments. My responses are as follows:
"In the paragraph on Growth Kinetics in vitro (322 to 351) and Figure 3. Please justify and discuss why TCID50 was used instead of GE/ml when assessing R120T mutants. Also, the difference between B and E needs better explanation. Particularly with respect to why the mutant didn't grow in the first experiment at 37C (B) as compared to the second experiment (E). How many replicates were performed? Where are the error bars? Were any statistics performed to assess differences in growth kinetics?"
I have inserted a clause (lines 334-335) reminding that both assays of genome equivalents and infectious virus were used alternatively to measure virus release into medium.
On lines 352-353, we comment on differences between the two experiments shown in Fig. 3B and 3E, and in 358-360, we explain that "We attribute the difference from results shown in Figure 3B to a random mutation that changed E123 to K123 and was thereafter selected for more rapid growth." which did not occur in Fig 3B. Although each experiment was repeated 2-3 times, the results shown represent a single replicate.
"Figure 4 figure legend indicates that significance was assessed using student's t test, but neither the figure or the discussion indicates that results of this analysis."
Unfortunately, when the editors of Viruses converted the figures I submitted to their desired format, the asterisks referred to in legend did not transfer, so I have deleted reference to the statistical analysis in the legend. I have resubmitted Figure 4 to see if the editors can do a more successful conversion.